# Endocrine Disruptor-Induced Bone Damage Due to Hormone Dysregulation: A Review

**DOI:** 10.3390/ijms24098263

**Published:** 2023-05-05

**Authors:** Nneamaka Iwobi, Nicole R. Sparks

**Affiliations:** 1Department of Pharmaceutical Sciences, School of Pharmacy, University of California, Irvine, CA 92697, USA; niwobi@uci.edu; 2Department of Occupational and Environmental Health, University of California, Irvine, CA 92697, USA

**Keywords:** osteoblasts, osteogenesis, hormones, endocrine toxicity, endocrine-disrupting chemicals, bone defects

## Abstract

Hormones are indispensable for bone development, growth, and maintenance. While many of the genes associated with osteogenesis are well established, it is the recent findings in endocrinology that are advancing the fields of bone biology and toxicology. Endocrine-disrupting chemicals (EDCs) are defined as chemicals that interfere with the function of the endocrine system. Here, we report recent discoveries describing key hormone pathways involved in osteogenesis and the EDCs that alter these pathways. EDCs can lead to bone morphological changes via altering hormone receptors, signaling pathways, and gene expression. The objective of this review is to highlight the recent discoveries of the harmful effects of environmental toxicants on bone formation and the pathways impacted. Understanding the mechanisms of how EDCs interfere with bone formation contributes to providing a comprehensive toxicological profile of a chemical.

## 1. Introduction

Congenital bone defects are a major public health concern. According to the World Health Organization (WHO) [1], birth defects are the second leading cause of deaths for infants (in 28 days) and children under 5 years of age, resulting in nearly 3.3 million deaths globally [1]. In the United States alone, 1 in every 33 babies are born with a birth defect each year, and accounts for 20% of infant and child mortality. This is a highly concerning issue that needs to be addressed. Congenital malformations can be attributed to genetic and non-genetic factors. Non-genetic factors, particularly environmental factors, are among the most concerning leading to an increased risk of birth defects [2,3]. Therefore, it is imperative to uncover the mechanisms by which environmental toxicants affect bone development and prompt the risk of skeletal defects. Osteogenesis is the process of bone formation, whereby osteoblasts, the bone forming cells, produce a mineralized extracellular matrix during early development, adult bone homeostasis, and bone remodeling after an injury [4]. Osteoblast lineage commitment is tightly controlled by mechanisms including epigenetic, transcription factors, and signaling pathways. Elucidating such genetic processes is key to understanding normal and abnormal bone development. Specifically, understanding how environmental factors contribute to the dysregulation of hormone signaling pathways during osteogenesis will help provide insights into the molecular mechanisms of bone disorders and diseases and the development of diagnostic tools and treatments.

This review brings forth recent information showing the effects EDCs on bone development and remodeling. Further, the review aims to identify the underlying mechanisms involved that are negatively impacted due to EDC exposure resulting in skeletal damage.

## 2. Method

A literature search was conducted on PubMed and Google Scholar for studies related to hormones and osteogenesis as shown in Table 1. Publications were screened for relevance to hormones role in osteogenesis and toxicant dysregulation.

## 3. Osteogenesis

Bone formation is the result of two processes: intramembranous ossification, which is the formation of flat bone, i.e., thin layers of connective tissue and top of the skull; and endochondral ossification, which is the process by which bone tissue, cartilage, is formed in early fetal development and then replaced with bone later [5,6,7]. Osteoblasts are derived from progenitor neural crest (NC) and mesodermal cells, where NC cells typically go through intramembranous ossification and endochondral ossification for mesoderm derived osteoblasts. A shared precursor between NC and mesoderm cells are mesenchymal stem cells (MSCs), which have the capacity to differentiate into osteoblast, chondrocytes, myoblasts, and adipocytes [8]. Proliferation, matrix maturation, and mineralization are the key stages of osteoblast development which require the expression of distinct osteoblast markers. The most common markers of osteoblast development are alkaline phosphatase (*ALP*), runt-related transcription factor 2 (*RUNX2*), type I collagen (*COL1A1*), osteopontin (*OPN*), bone sialoprotein (*BSP*), and osteocalcin (*OCN*). *ALP*, *RUNX2*, and *COL1A1*, which are early osteoblast markers, and *OPN*, *BSP*, and *OCN* represent later stages of osteoblast differentiation [9,10,11,12]. Exposure to environmental toxicants, such as air pollution, flame retardants, or tobacco products, during these susceptible periods of development can lead to unwanted life-long bone defects, diseases, and disorders [13]. Therefore, it is necessary to understand how exposure can impact the mechanisms of bone development and result in the developmental toxicity of bone.

## 4. Osteogenesis and Its Hormone Regulation

Hormones are major contributors to osteogenesis and deviations in hormone expression can lead to an undesired bone formation outcome. Bone development is hormone dependent, with each hormone having its own receptor in bone tissue and controlled by several endocrine glands [14]. Therefore, these pathways are susceptible to endocrine disruption by environmental insults, including EDCs, that can cause osteogenic defects.

**Thyroid hormone.** Thyroid signaling plays an important role in many cells within the human body and is involved in metabolism maintenance as well as body growth and development [15,16]. Thyroid hormones (TH) aid in osteoblast formation in the early stages of skeletal development, as well as bone growth and maturation. There are three subtypes make up thyroid receptors (TRs): TRα1, TRβ1, and TRβ2, where TRα1 and TRβ1 are most expressed in bone [17]. Thyroid hormones positively regulate osteoblast differentiation via bone morphogenetic protein (BMP) and IGF1 signaling as seen in Figure 1. Positive osteoblast development is supported through the BMP/SMAD signaling pathway observed in mouse osteoblasts treated with TH (T3). Hormones triidothyronin (T3) and thyroxine (T4) are the two main forms of TH, where T4 is the primary form. Secondary, T3 is produced through the enzymatic conversion of T4 [15,16,17]. T3 led to BMP activation and SMAD1/5/8 phosphorylation that yielded enhanced osteoblast differentiation potential [18]. In differentiating MC3T3-E1 pre-osteoblast cells, T3 and T4 treatments increased *Igf-1* mRNA levels supporting osteoblast differentiation [19]. TH has been shown to regulate osteoblast differentiation through WNT/β-catenin signaling pathway stimulation or inhibition. The crosstalk between THs and WNT signaling needs to be fully delineated in bone compared to more established mechanisms in other tissues [20]. When treated with T3, WNT signaling activity was decreased in mouse osteoblast cells [21]. In vivo, β-catenin levels were stabilized with a mutant thyroid hormone receptor to activate WNT signaling in the presence of TH and increase bone mass [21]. In contrast, Tsourdi et al. [22] found WNT signaling inhibitor DKK1 serum levels were increased in hypothyroid mice, which correlated with decreased bone formation [22]. In addition, BMP signaling can regulate WNT/β-catenin signaling to regulate osteoblast differentiation and bone formation [23].

**Parathyroid hormone.** The parathyroid hormone (PTH) is an 84-amino acid peptide hormone secreted by the parathyroid glands. PTH mainly acts on the bone and kidney. It is crucial for osteoblast differentiation and post-natal bone calcium and phosphorus maintenance. PTH-related protein (PTHrP) is crucial for endochondral bone formation during pre- and post-natal bone formation [24]. PTH and PTHrP are similar peptide hormones that share interaction with a single common receptor, PTH type I receptor (PTH1R), predominantly through cyclic adenosine monophosphate/protein kinase A (cAMP) [24]. These receptors are found in progenitor and osteoblast cells. Figure 2 demonstrates PTH stimulation of osteoblast development mediated through the cyclic AMP and BMP signaling pathways downstream of the PTH1R [24,25,26,27,28]. PTH-induced BMP signaling stimulation phosphorylates SMAD1, which prevents the inhibitory effect of NOGGIN and increases the endocytosis of PTH/PTH1R/LRP6, which induces β-catenin stabilization. Increased PTH enhances MSC differentiation into osteoblasts through BMP signaling [29]. PTH and PTHrP stimulate pro-osteogenic genes, *RUNX2*, *ALP*, and *OCN*. Expressed at the correct timing of development, PTH increases osteoblast differentiation. PTH receptor (PTHR) deletion in bone marrow cells resulted in an increase in bone marrow adiposity and bone resorption, along with a physically visible low bone mass in mice [30].

**Vitamin D.** Cells of the osteoblast lineage are responsive to systemic hormones such as 1,25-dihydroxyvitamin D_3_ (1,25(OH)_2_D_3_). Vitamin D is a steroid hormone with an essential role in bone metabolism. The active form of vitamin D, 1,25(OH)_2_D_3_, binds to the vitamin D receptor (VDR), which heterodimerizes with the retinoic X receptor (RXR) and activates target genes. Increased vitamin D levels enhance bone formation by promoting osteoblast differentiation and mineralization [31], provided in Figure 3. Mouse overexpression of the human *VDR* gene increased cortical and trabecular bone supporting 1,25(OH)_2_D_3_ impact on bone. Similarly, in antigen-induced arthritis (AIA) rats that have significant bone loss, 1,25(OH)_2_D_3_ treatment increased trabecular bone volume compared to untreated AIA rats and healthy control rats [32]. However, the 1,25(OH)_2_D_3_ administration did not have any anti-inflammatory effect. MSCs treated with exogenous 1,25(OH)_2_D_3_ differentiate into osteoblasts that produce a mineralized extracellular matrix that enhanced differentiation. Cell culture medium supplementation with 1,25(OH)_2_D_3_ triggers human embryonic and induces pluripotent stem cell osteoblast differentiation [33,34]. Human MSC and mouse embryonic stem cell studies resemble human pluripotent stem cell studies showing increased osteoblast differentiation with 1,25(OH)_2_D_3_ treatment [35,36].

**Estrogen.** Estrogen is a key hormone involved in the development and homeostasis of bone tissue in both males and females. Estradiol is the most potent estrogenic hormone in the human body. Estrogen action is controlled by two main estrogen receptors (ER), alpha and beta (ERα and ERβ), encoded by *ESR1* and *ESR2*, respectively. It regulates gene expression, metabolism, cell growth, and proliferation by acting through cytoplasmic signaling pathways or activating transcription in the nucleus, seen in Figure 4A. Estrogens bind to their receptors in the nucleus, acting as transcription factors regulating the expression of target genes. Estrogens can also bind to their receptors outside of the nucleus activating signaling pathways in the cytoplasm. The cytoplasmic signaling pathway is activated by estrogen and growth factors and acts though the kinase signaling cascade which phosphorylates substrate proteins and transcription factors [37,38]. Estrogen treatment has been found to induce osteoblast differentiation and activate ERK/JNK signaling, cell cycle regulation, cell growth, and the survival pathway in rat bone marrow-MSCs. In the WNT pathway, activation of ER signaling induces osteogenic differentiation and matrix mineralization [39,40,41]. A deficiency in estrogen is associated with reduced bone formation. Estrogen prevents bone loss by inhibiting osteoclast—the bone-resorbing cell—activity. *Esr1* deletion in female mice osteoclasts resulted in increased osteoclast numbers and reduced trabecular bone mass [42]. Nakamura et al. [42] concluded that estrogen’s osteoprotective effect was through the expression of Fas ligand (*FasL*) in osteoblasts that induced osteoclast apoptosis, as depicted in Figure 4B [42]. Another mechanism of estrogen-mediated osteoclast inhibition involves the receptor activator of nuclear factor κB ligand (*RANKL*) regulation [42]. RANKL is essential for osteoclast differentiation and can be suppressed by osteoprotegerin (*OPG*). In estrogen deficient C57BL/6 mice, increased bone resorption activity was found due to the lack of ERα-mediated suppression of *Rankl* expression in bone lining cells, which RANKL binds to RANK on the surface of osteoclast progenitors to initiate the bone breakdown [43]. In addition, estrogen deficiency has been linked to oxidative stress and inflammation, which can increase bone resorption [44].

## 5. Mechanisms of Endocrine Disruption and Bone Damage

In the past two decades, globally, considerable efforts have been made in assessing endocrine disruption/dysfunction and the adverse effects on tissues, particularly of liver, kidney, heart, bone, and the nervous system. Environmental chemicals can mimic or interfere with the body’s hormones. EDCs and have been linked to poor reproductive, developmental, neurogenic, and osteogenic outcomes [45]. EDCs are found in common everyday products, such as plastics, metal food cans, toys, detergents, flame retardants, and air pollution; thus, endocrine toxicology has become an important field of research. Animal and epidemiological studies support the findings of adverse effects of EDCs on bone formation. It is important to understand the molecular mechanisms of how chemicals can exert effects on hormone systems resulting in adverse effects on bone development and maintenance [46].

**Tobacco.** Tobacco is a key player in various health complications, including chronic diseases, cancers, reproductive and developmental disorders, and premature death [47]. It has also been confirmed that tobacco smoking contributes to bone defects, including imbalanced bone turnover and delayed healing after injury, which, in turn, results in both low bone mass and bone mineral density. Smoking is a strong lifestyle risk factor for osteoporosis and bone fracture [48]. Cross-sectional studies suggested that smoking enhances bone loss, where postmenopausal smoking women had decreased bone density compared to nonsmokers, which continued to decline overtime [49]. The association between tobacco smoking, hormone system dysregulation, and bone defects has been reported. Studies have shown that exposure to cigarette smoke has altered estrogen levels within women, leading to fertility problems and early menopause [50,51,52]. In addition, tobacco also impacts estrogen by preventing the conversion of androgen to estrogen by inhibiting aromatase [53]. Vitamin D levels are often reduced by the introduction of tobacco into the body [54]. There is a negative correlation between vitamin D levels and cigarette smoking in humans [55]. In vitro studies have suggested a link between tobacco use and WNT signaling inhibition, reporting that smoking increases *DKK1* levels and reduces bone formation [56], and further, showing that WNT inhibition caused by cigarette smoking induced osteoclast differentiation via increased *RANKL* expression [56]. Though more studies are needed to understand the mechanisms of toxicity, zebrafish embryos’ exposure to cigarette smoke was shown to be developmentally toxic to the craniofacial skeleton [57].

**Air pollution.** Air pollution was estimated to cause 100,000 and 4.2 million premature deaths annually in the United States and worldwide, respectively. These deaths are related to respiratory disease, cancers, cardiovascular disease, adverse birth outcomes, and other health impacts [58]. Worldwide, air pollution was the second largest risk for noncommunicable diseases and the largest environmental health risk [59,60]. A major contributor to air pollution is particulate matter (PM), particles (solid or liquid) suspended in air and defined by their size that are typically attributed to the health effects of air pollution [59,61,62,63]. Air pollution is highly concerning because of how prevalent it is in our everyday lives and is a major contributing factor to multiple diseases and cancers, which can result in premature deaths [64]. Air pollution has been linked to decreased bone mineral density, increased risk of bone fracture, and effects on the hormonal routes involved with osteogenesis. Studies have shown postmenopausal women with decreased estrogen levels and increased *RANKL* expression associating to bone loss [59]. There is evidence of a negative correlation between the air pollutant indicators, PM_2.5_ and PM_10_, and bone mineral density [65], emphasizing the impact of air pollution as a risk of poor bone health. PM exposure can lead to a vitamin D deficiency, inhibiting conversion of inactive vitamin D to active 1,25(OH)_2_D_3_ [59]. Cross-sectional analysis suggested PM exposure decreased bone mineral content through PTH impairment [61].

**Flame Retardants.** Flame retardants are used in a multitude of everyday items, including furniture, carpets, and electronics to prevent combustion and reduce the risk of fire. Traditional flame retardants, such as polybrominated diphenyl ethers (PBDEs), are being replaced with organophosphorus flame retardants (OPFRs). Two globally used OPFRs are Tris (1,3-dichloroisopropyl) phosphate (TDCIPP) and triphenyl phosphate (TPhP) [66]. OPFRs have been revealed to be developmentally toxic and can disrupt the endocrine system, where changes in TH levels are associated with developmental neurotoxicity [67]. Medaka *Oryzias melastigma* toxicology studies reported both TDCIPP and TPhP are bone developmental toxicants. Medaka exposed to TDCIPP and TPhP presented malformed pectoral fins, reduced body length, and curvature of the spine. TDCIPP and TPhP induced bone developmental toxicity through the misregulation of *bmp* and *runx2* [66]. The endocrine-disrupting capability of TDCIPP and TPhP requires further exploration, but a few reports suggest that flame retardants mimic estrogen and inhibit estrogen sulfotransferase, altering estrogen metabolism, and thus eliciting endocrine disruption [68].

**Pesticides**. Common household products, pesticides, are a threat to human health and wildlife, and many have been identified as endocrine disruptors. Pesticides are hazardous chemicals used to prevent, destroy, repel, or mitigate organisms, including insects, rodents, and plants [69]. Due to its increasing usage and persistence in the environment, humans are exposed to pesticides through their diet, environment, and occupation that can be harmful. Prolonged pesticide exposure can result in organ damage/failure, reproductive issues birth defects, and various cognitive impairments [70,71]. Pesticides can disrupt hormones of the human body’ by binding to and activating or inactivating major receptors, such as ER and androgen receptor, they disrupt hormone synthesis, metabolism, and natural hormone levels [72,73,74,75]. Organophosphate pesticides, including dichlorvos, dimethoate, acephate, and phorate, have been associated with bone loss. This bone loss is due to kidney dysfunction and obstruction of vitamin D to 1,25(OH)_2_D_3_ conversion [76]. Chronic kidney disease individuals often have increased risk of low bone mineral disorders and bone fractures [77].

Dichlorodiphenyltrichloroethane (DDT) is a common pesticide that can alter natural hormones levels [78]. Evidence points towards DDT mimicking estrogenic effects in vitro, where DDT-treated MSC cells displayed ER binding and led to increased estrogen levels, as with cells treated with estrogen alone [79]. DDT exposure changed the MSCs’ global gene profiles and natural self-renewal, proliferation, and differentiation ability. DDT’s metabolite, Dichlorodiphenyldichloroethylene (DDE), is a persistent metabolite; and exposure can induce vomiting, nausea, tumors, compromised immune systems, and increased chance of preterm birth [80,81,82]. High levels of DDE have been correlated with bone marrow defects, where bone marrow HL-60 cells had decreased cell viability, cell morphology abnormalities, hindered cell differentiation, and increased Ca2+ [83]. DDT exposure can cause thyroid dysfunction, which leads to bone loss [76]. However, DDT’s effects and mechanisms on hormones and bone disorders have not yet been fully determined. In vivo studies of imidacloprid, a neonicotinoid insecticide, have revealed that the insecticide can change hormone levels, promote DNA damage, attack reproductive organs, disrupt in utero development, and bone mineral composition [84,85,86,87]. When examining the effects of imidacloprid on bone development, in vitro chicken embryo studies revealed that early exposure to the pesticide results in cranial bone defects of the embryo. The study reported cranial NC cell differentiation. The inhibition can be correlated to suppressed *Msx1* and *Bmp4* expression and therefore attenuating osteogenesis [88].

**Para-Nonylphenol (P-NP).** P-NP is representative alkylphenol widely used in detergents, emulsifiers, and solubilizers, which can accumulate in the environment and has been known to reduce osteogenesis and cell viability through inducing apoptosis in osteoblast cells. In vitro studies revealed that MSC exposure to low-levels of p-NP inhibited osteoblast mineralization compared to controls. In addition, the MSCs had increased apoptosis, observed by chromatin condensation and nuclear breakage, along with cytoplasm shrinkage [89]. Rat MSCs differentiating into osteoblasts had diminished mineralization when exposed to 2.5 μM p-NP and reduced expression of osteogenic genes *Alp*, *Smad*, *Bmp*, and *Runx2*. In addition, the MSCs showed metabolic imbalance and oxidative stress [90]. Studies reporting the mechanisms of p-NP endocrine toxicity in bone are lacking. Reports do suggest that p-NP acts with estrogen-like activity [91]. Since bone is an estrogen target that expresses both ERα and ERβ, it is plausible that p-NP can decrease bone formation. Similarly, p-NP is toxic to the thyroid gland, where pregnant dams exposed to p-NP had decreased T3 and T4 serum levels [92]. The study found that p-NP affected litter size, body weight, and tail length but did not report specific findings about the skeleton.

**Bisphenol A (BPA).** BPA is a chemical found in nature and a major component of epoxy and polystyrene resins and polycarbonate plastics. Bisphenols can imitate or block hormone receptors that have been linked to various health complications [93]. BPA and its analogs (BPF, BPS, and BPAF) adversely affect osteogenic gene expression in human osteoblasts. The analogs deregulated *ALP*, *COL1A1*, and *OCN* expression, inhibiting matrix formation and mineralization, similar to the effects of BPA [94]. BPA can alter bone formation by mimicking estrogen and competing for both ERα and ERβ. Female rat offspring exposed to 10 μg/kg of BPA per day during gestational days 14–21 had delayed bone development and reduced bone mass. In vitro findings attributed attenuated osteogenesis via ERβ downregulation [95]. BPA has been discovered to reduce the levels of vitamin D. Pregnant women showed a negative correlation between vitamin D levels and BPA exposure [96]. In an elderly sample population, a cross-sectional study over one year reported an inverse relationship between urinary BPA concentrations and vitamin D serum levels. The study suggests that the low vitamin d levels are due to inhibited 1,25(OH)_2_D_3_ synthesis and the population is at risk for bone-related diseases [97].

**Perfluoroalkyl and polyfluoroalkyl substances (PFAS).** PFAS are a large group of chemicals that have been in use since the 1950s and are persistent environmental toxicants due to their stability and presence in items we use every day. PFAS have been classified as EDCs and many studies are reporting their toxicological effects. Primary exposure to PFAS occurs through food and drinking water, where fish are a major contributor. After exposure, PFAS can accumulate in multiple tissues, including bone, and alter bone development [50,98,99,100,101]. However, the mechanisms associated with PFAS bone toxicity are underexplored. From bone banks and cadaver samples, human trabecular and bone marrow contained PFAS [99]. The potential consequence of PFAS accumulation is increased bone resorption activity seen in human osteoclasts differentiated from bone marrow and peripheral blood samples [99]. PFAS treatment in Jeg-3 cells suggests WNT/β-catenin suppression as a potential mechanism [98]. In vitro and in silico studies reported PFAS binding to the VDR. PFAS’s competition for VDR binding resulted in reduced target gene expression and osteoblast differentiation mineralization [101]. Epidemiological studies suggest women are more responsive to PFAS compared to men. PFAS exposure was linked to delayed puberty, early menopause, and low estradiol concentration, which correlated to low bone mineral density and microarchitectural changes [50].

**2,3,7,8-tetrachlorodibenzo-p-dioxin (TCDD).** TCDD is a highly toxic chemical which has been shown to disrupt hormone signaling. Exposure to TCDD is developmentally toxic to bone and can cause craniofacial defects. TCDD inhibited human fetal palate mesenchymal cells’ (hFPMCs) osteoblast differentiation [102]. The hFPMC differentiation inhibition revealed that TCDD inhibited BMP/SMAD signaling and therefore promoted osteoblast decline through decreased cell proliferation, ALP activity, and calcium deposition. The aryl hydrocarbon receptor (AhR), involved in osteogenesis, regulates the synthesis and metabolism of estrogen in bone tissue [103]. TCDD has a high binding affinity to AhR that attenuates osteogenic transcriptional regulation, osteogenic differentiation, tibial growth, and bone mineral density [14,102,103,104,105,106]. TCDD at 10 nM disrupted human bone marrow MSC differentiation via downregulation of osteogenic markers *ALP*, *DLX5*, and *OPN* [101]. Human MSC osteogenesis was rescued with coadministration of TCDD and AhR antagonist GNF351 to block TCDD. TCDD has been found to decrease TH levels, T4, and thyroid function. However, more studies are needed for better understanding of the mechanism of TCDD toxicity on bone through thyroid hormone signaling [107].

## 6. Conclusions

The literature has established that hormones are key players in osteogenesis and that their misexpression can cause differentiation inhibition, porous bone, and bone loss; however, research about the mechanisms associated with bone loss and environmental influences is lacking. This review aimed to shed light on current research about hormones and osteogenesis, linking environmental-mediated endocrine disruption to skeletal diseases and disorders. It should be noted that there are many hormones in the body, including growth hormones, insulin, and follicle stimulating hormones, that were not addressed in this review; the authors selected four hormones and discussed how they are influenced by toxicant exposure. Hormones play a large role in the development of osteoblasts and bone formation, supporting osteogenesis through genes and/or signaling that are/is pro-osteoblast or osteoclast inhibitory. Our understanding of how environmental chemicals, such as EDCs, can disrupt the molecular mechanisms of bone development has seen recent progress, but more toxicological studies linking EDC exposure to hormone signaling impairment and defected bone differentiation and formation are needed. So far, the effects of environmental EDCs on hormone signaling resulted in reduced osteoblast differentiation and adult bone mineral density due to increased osteoclast activity and changes in the WNT, BMP, and vitamin D pathways (Figure 5). These studies primarily focused on epidemiology data and adult bone formation using adult bone tissue and MSCs, excluding pre- and post-natal bone development. Pre-natal, childhood, and adolescent periods should be deemed as susceptible periods of development, vulnerable to toxicant exposure. Therefore, we need an increase in studies investigating the impact of endocrine disruptors on pre-natal and post-natal bone development. Further studies are needed to explore the toxicological and mechanistic implications of EDC-related hormone disruption and to identify hormone-disrupting toxicants as risk factors for skeletal diseases and disorders.

## 7. Future Directions

To date, studies have shown that environmental chemicals, including EDCs, interfere in the viability, differentiation, and function of osteoblasts and osteoclasts via hormone signaling impairment, resulting in bone loss. Embryonic and early childhood exposure to environmental factors that can cause birth defects are poorly understood and of great interest. It is specifically the periods of rapid cell division and differentiation that make pregnancy and early childhood susceptible to environmental factors that can exert an adverse influence on human bone development. Future research efforts will need to work towards understanding how endocrine disruptors affect pre- and post-natal bone development; specifically, by teasing out differences between neural- and mesoderm-derived bone responses to EDCs. This information will provide the scientific community with a mechanistic understanding of endocrine and bone toxicity and will assist the development of strategies for endocrine toxicity assessment, diagnostic tools, such as biomarkers of osteotoxicity, and therapeutic strategies.

## Figures and Tables

**Figure 1 ijms-24-08263-f001:**
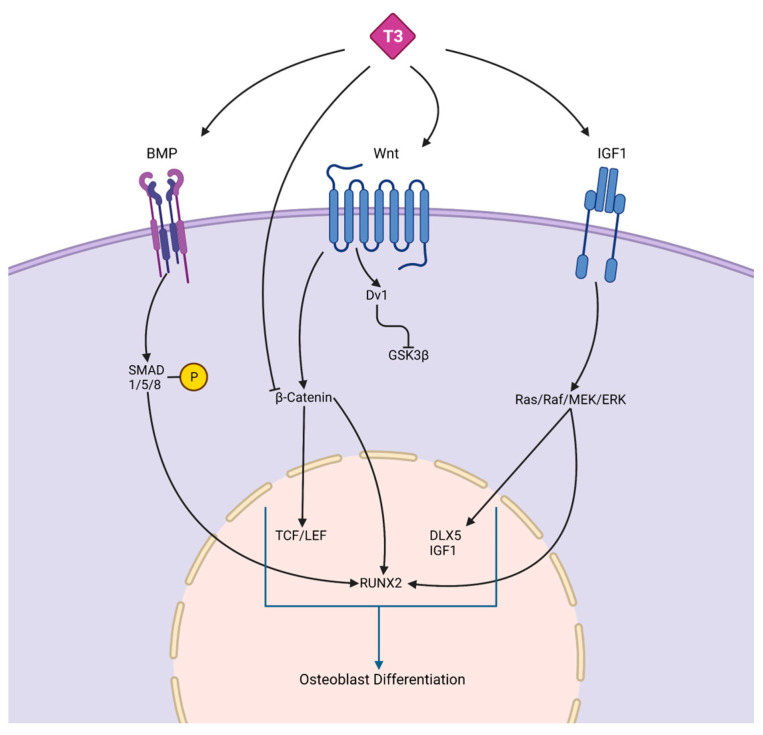
Crosstalk between thyroid hormones and signaling pathways. The schematic depicts the complex interaction between thyroid hormones, such as T3, and BMP, WNT, and IGF1 signaling, which are responsible for osteoblast differentiation. BMPs bind to receptors on osteoblast progenitors to activate SMADS, leading to increased *RUNX2*. *RUNX2* is an osteogenesis specific transcription factor that promotes osteogenic related genes expression. In the WNT/β-catenin pathway, TH regulates osteoblast differentiation through either inhibiting β-catenin, which prevents osteoblast differentiation, or binding to WNT, which promotes osteoblast differentiation through accumulating β-catenin, increasing the levels of *TCF/LEF* and *RUNX2*. IGF-1 receptor-induced osteogenesis activates the Ras/Raf/MEK/ERK pathway, leading to an increase in osteogenic genes.

**Figure 2 ijms-24-08263-f002:**
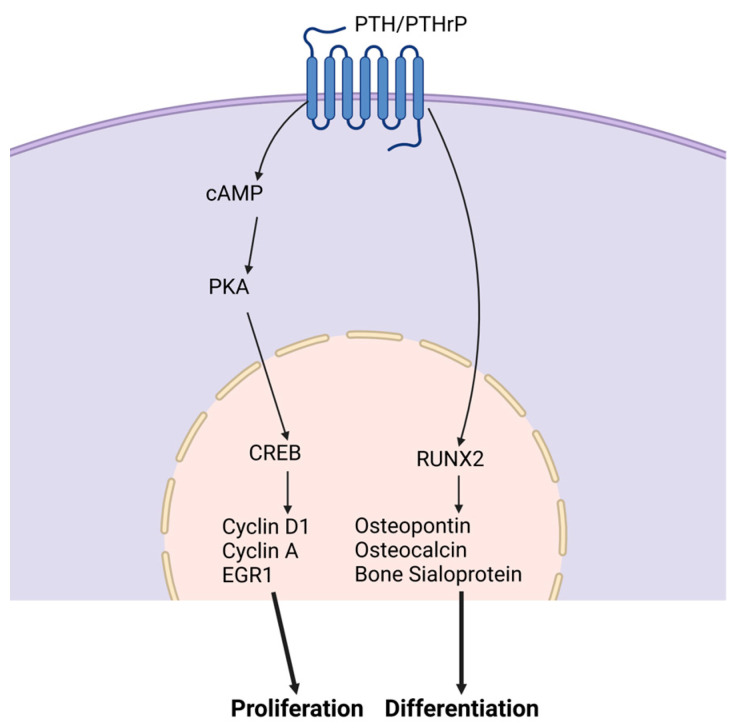
PTH and PTHrP signaling pathways. PTH and PTHR stimulate the proliferation and differentiation of osteoblasts. To promote proliferation, cAMP is activated followed by an increase in PKA levels. These cellular outcomes are mediated through elevation of intracellular cAMP via the PTH receptor. This increase leads to the activation of CREB in osteogenic cells.

**Figure 3 ijms-24-08263-f003:**
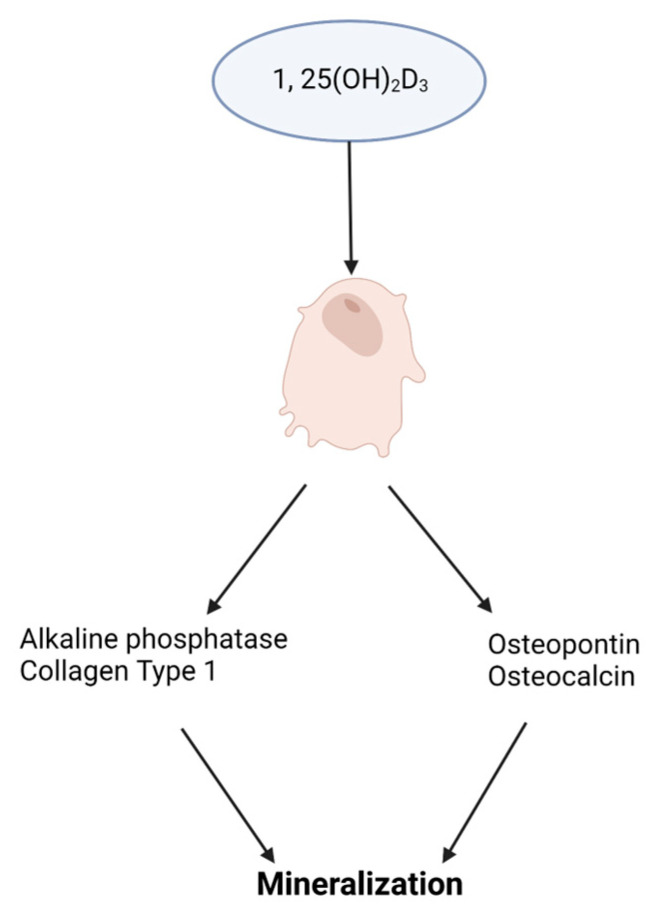
Impact of vitamin D3, 1,25(OH)_2_D_3_, on osteoblast differentiation. A schematic illustrating 1,25(OH)_2_D_3_ stimulating the expression of osteoblast promoting genes for extracellular matrix mineralization.

**Figure 4 ijms-24-08263-f004:**
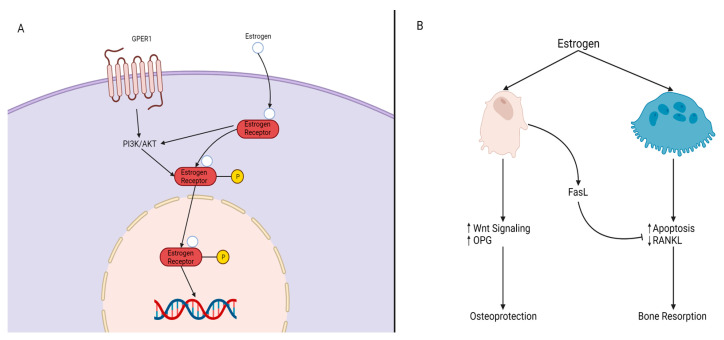
Mechanisms of Estrogen Signaling. (**A**) P13K/AKT is impacted by either G protein-coupled estrogen receptor, GPER, or by the estrogen/estrogen receptor (ER) complex, resulting in the phosphorylation of the estrogen/ER complex. The complex then crosses the nucleus and elicits its response on target genes. (**B**) Estrogen impacts both osteoblasts and osteoclasts. In the presence of estrogen, osteoblasts experience an increase of WNT signaling and *OPG* levels and produce *FasL*. *FasL* inhibits osteoclast activity through reduced RANKL expression and osteoclast apoptosis, resulting in osteoblast protection and maintenance (osteoprotection). The absence of estrogen leads to bone resorption.

**Figure 5 ijms-24-08263-f005:**
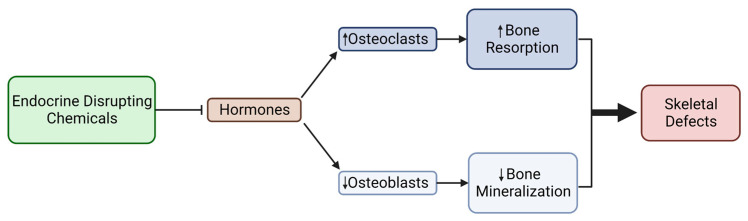
Endocrine-disrupting chemicals’ attenuation of bone formation mediated through hormone signaling changes, resulting in defects of the skeleton.

**Table 1 ijms-24-08263-t001:** Literature search and selection criteria.

	Literature Search Strategy
A.	PubMed and Google Scholar combinations (name of hormone + bone-related term or toxicant + hormone + bone-related term) of the following terms: hormones, thyroid hormone, parathyroid hormone, vitamin D, estrogen, osteogenesis, bone, bone development, bone damage, bone birth defects, bone toxicology, endocrine disruptors, endocrine-disrupting chemicals, tobacco, cigarette smoke, air pollution, flame retardants, bisphenol A, PFAS, TCDD, DDT, para-nonylphenol, and pesticides.
B.	English language papers were screened from publication date 1 January 2000 to 30 April 2023. Publications included original research articles, reviews, and book chapters consisting of in vitro studies, in vivo animal studies, human studies, and meta-analyses. Non-English and unavailable full-text articles were excluded.

## Data Availability

Not applicable.

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
