# Peer review of "Endocrine Disruptor-Induced Bone Damage Due to Hormone Dysregulation: A Review"

_ijms, 2023, doi:10.3390/ijms24098263_

Round 1
Reviewer 1 Report
Dear Authors,
The article is very interesting. The topic of endocrine disruptors targeting, among others, the skeleton status, is very modern; we continuously need to update this domain due to the advance of technology, industry procedures and technologies targeting overall health. Particularly, bone field remains a major area of interest due to massive epidemiologic impact of primary (post - menopausal and age – related) osteoporosis all over the world which seems to associate an increasing prevalence due to aging population and due to still being under diagnosed in certain geographic regions.
I enjoyed reading the article which I consider to bring value to our knowledge by its complexity and novelty.
I only have a few minor observations:
1. Title - I suggest not to repeat “bone”, for instance “Hormone regulation of bone development: an overview of endocrine disruptors and skeleton damage”.
2. Abstract Line 16 – Please specify “Our aim (or objective) is to highlight recent..:
3. Introduction – The data from first three lines need a specific list of references (concerning the numbers you cited), and this is different from pathogenic (genetic and toxic elements, etc.) issues you mentioned at lines 29-30.
4. At the end of subsection 1 (Introduction) please introduce a statement concerning the specific aim (objective or endpoint) of the current work.
5. After introduction, a section dedicated to Methods is suggested (highlighting the criteria of selection with regard to the papers you cited) and the type of the article (narrative review). It is important to specify if you choose articles with clinical evidence, experimental data, murine studies since they are many levels of intervention, and some are not completely proved in humans.
6. Section 3 – Line 79. Please use “thyroid hormones” instead of “thyroid hormone” since they are two: T4 and T3.
7. Please explain T4 (levothyroxine) and T3 (triiodothyronine) when first used.
8. After mentioning each name of an author/researcher within main text, the citation number should be introduced again in addition to the citation at the end of the statement. For example, “Tsourdi et al. [21] found WNT…[21].”
9. I suggest to separate the actions of PTH from PTHrP since they are distinct in different stages of life (for instance, the role of PTHrP during fetal development) (lines 102-114)
10. Each subsection starts with the name of the hormone at the beginning of a statement with the exception of vitamin D. Maybe you use a uniform style. The same observation is for line 220 (you repeated “flame retardants”).
11. Testosterone, glucocorticoids and growth hormones, and, also, gut - originating serotonin are key players within bone metabolism.
12. Discussion – a section dedicated to current limits of the topic and further expansion of the studies to increase the level of clinical evidence is advised.
13. Conclusions. Please provide a clear take home message and avoid references at this section.
Thank you
Dear Editor,
The article is very interesting. I introduced my observations to the authors.
Thank you very much,
Best regards,
Author Response
"Please see the attachment."

Reviewer 2 Report
The manuscript “Hormone Regulation of Bone Development: An Overview of Endocrine Disruptors and Bone Damage” present and interesting review that can be published after minor corrections.
The authors should include, in the PTH section, interesting study describing microbiota effect on the bone response to PTH. (Pacifi R)
As estrogens is a major regulator of bone homeostasis and that their withdrawal induce inflammation, a comment about the relation between inflammation and bone damage should also be included in the estrogen section.
Author Response
"Please see the attachment."
